# Optimized and Validated Stability-Indicating RP-HPLC Method for Comprehensive Profiling of Process-Related Impurities and Stress-Induced Degradation Products in Rivaroxaban (XARELTO)^®^

**DOI:** 10.3390/ijms26104744

**Published:** 2025-05-15

**Authors:** Aktham H. Mestareehi

**Affiliations:** 1Department of Applied Pharmaceutical Sciences and Clinical Pharmacy, Faculty of Pharmacy, Isra University, P.O. Box 22, Amman 11622, Jordan; 2Department of Pharmaceutical Sciences, Eugene Applebaum College of Pharmacy and Health Sciences, Wayne State University, Detroit, MI 48202, USA; aktham.mestareehi@med.wayne.edu; 3Department of Pharmaceutical Sciences, School of Pharmacy, Northeastern Illinois University, Chicago, IL 60625, USA

**Keywords:** rivaroxaban, LC-PDA, specificity, linearity, accuracy, precision, limit of detection (LOD), limit of quantitation (LOQ), stress-induced degradation, impurity

## Abstract

An isocratic reverse-phase high-performance liquid chromatography (RP-HPLC) method, coupled with photodiode array detection (PDA), was developed for the identification and characterization of stress degradation products and an unknown process-related impurity of rivaroxaban in bulk drug form. Rivaroxaban, a selective and direct Factor Xa inhibitor, underwent forced degradation under hydrolytic (acidic, alkaline, and neutral), photolytic, thermal, and oxidative stress conditions, following the ICH’s guidelines. The drug displayed significant susceptibility to acid, base, and oxidative environments leading to the formation of eleven degradation products. All degradation products, along with process impurities and Rivaroxaban, were effectively separated using a (4.6 × 250 mm, 5 µm) C18 Thermo ODS Hypersil column at ambient temperature. The mobile phase composed of acetonitrile and monobasic potassium phosphate (pH 2.9) in a 30:70 (*v*/*v*) ratio, with a flow rate of 1.0 mL/min, and detection was carried out at 249 nm. The LC-PDA method was validated in accordance with the ICH’s guidelines and USP38-NF33, demonstrating specificity, linearity, accuracy, precision, and robustness. Recovery studies showed results within the range of 98.6–103.4%, with a % RSD LT 2%. The limits of detection (LOD) and quantitation (LOQ) for rivaroxaban were determined to be 0.30 ppm and 1.0 ppm, respectively. Stress studies confirmed that the degradation products did not interfere with rivaroxaban detection, establishing the method as stability-indicating. Specific impurities were identified, including impurity G at 2.79 min, impurity D at 3.50 min, impurity H at 5.32 min, impurity C at 6.14 min, impurity E at 8.36 min, impurity A at 9.03 min, and impurity F at 9.49 min. Additionally, several unknown impurities were observed at 3.20, 4.00, 4.59, and 4.77 min. Statistical evaluation confirmed the method’s reliability, making it suitable for routine analysis, quality control of raw materials, formulations of varying strengths, dissolution studies, and bioequivalence assessments of rivaroxaban formulations.

## 1. Introduction

Rivaroxaban is an oral anticoagulant based on the oxazolidinone structure, designed to be a selective and direct inhibitor of Factor Xa. It is routinely prescribed to prevent venous thromboembolism in adult patients undergoing procedures such as hip- or total-knee replacement surgery [1]. Rivaroxaban, originally developed by Bayer, is commercially marketed in the United States by Janssen Pharmaceuticals, a subsidiary of Johnson & Johnson. Notably, rivaroxaban holds the distinction of being the first direct oral Factor Xa inhibitor approved for clinical use [2]. It is quickly absorbed following oral administration, with peak plasma levels typically reached within 2 to 4 h. The drug demonstrates excellent oral bioavailability—approximately 80% to 100% for the 10 mg dose, regardless of food intake—and retains similarly high bioavailability for the 15 mg and 20 mg doses when taken with food. The synthetic pathway for rivaroxaban employs (R)-epichlorohydrin as a crucial chiral intermediate, which reacts with sodium cyanate (NaOCN) to yield (R)-chloromethyl-2-oxazolidinone. Bromobenzene serves as the major starting material in this synthesis route [3].

Rivaroxaban, the active compound in XARELTO^®^ tablets, has the chemical formula C_19_H_18_ClN_3_O_5_S and a molecular mass of 435.89 g/mol. Its structural formula is illustrated in Figure 1, with a reported purity of ≥99%. The chemical name of rivaroxaban is (5-chloro-N-({(5S)-2-oxo-3-[4-(3-oxomorpholin-4-yl)phenyl]-1,3-oxazolidin-5-yl}methyl)thiophene-2-carboxamide) [2,4]. It is an odorless, non-hygroscopic, and white to yellowish powder, and it exists as a pure (S)-enantiomer. Rivaroxaban has a melting point of 228–229 °C and a boiling point of approximately 732.6 ± 60.0 °C. Its density is 1.460 g/cm^3^, and it exhibits an optical rotation [α]D of −34 to −44 (c = 0.3 in DMSO), with a pKa of 13.36 [5]. Rivaroxaban is classified under the Biopharmaceutical Classification System as a low-solubility, high-permeability compound (Class 2). It has limited pH-independent solubility in aqueous media (5–7 mg/L; pH 1–9) but is slightly soluble in polyethylene glycol 400 (2431 mg/L) [3]. Each XARELTO^®^ tablet contains 20 mg, 15 mg, or 10 mg of rivaroxaban as the active ingredient. The formulation includes inactive excipients, such as magnesium stearate, hypromellose, lactose monohydrate, microcrystalline cellulose sodium lauryl sulfate, and croscarmellose sodium. Each tablet is film-coated, with the coating’s composition specifically tailored to correspond to the respective dosage strength [1].

Xarelto^®^ (rivaroxaban) is an oral anticoagulant that exerts its therapeutic effect through direct inhibition of Factor Xa, a critical enzyme in the coagulation cascade. By blocking Factor Xa activity, rivaroxaban prevents the conversion of prothrombin to thrombin, thereby disrupting fibrin formation and, ultimately, inhibiting blood clot development. Unlike traditional anticoagulants, which rely on cofactors such as antithrombin III, Xarelto is a highly specific inhibitor of Factor Xa. This mechanism of action underpins its efficacy in the prevention and treatment of thromboembolic disorders, including deep vein thrombosis (DVT), pulmonary embolism (PE), and stroke prevention in patients with non-valvular atrial fibrillation [1,3].

Xarelto (rivaroxaban) should be withheld for a minimum of 24 h prior to any surgical procedure to reduce the risk of bleeding. Discontinuing Xarelto prior to surgery minimizes the potential for excessive bleeding during the operation, given its anticoagulant properties [2]. Post-procedure, Xarelto should be resumed as soon as clinically appropriate, considering the patient’s condition and the type of surgery performed. If oral administration of Xarelto is not feasible after surgery, a parenteral anticoagulant (e.g., heparin) may be used until oral anticoagulation can be reinitiated. This strategy provides ongoing thromboprophylaxis while minimizing the risk of bleeding complications [6].

An excessive dose of Xarelto (rivaroxaban) may lead to bleeding complications due to its anticoagulant action. However, when doses exceed 50 mg in a single administration, systemic absorption does not significantly increase, as the drug’s absorption becomes saturated at higher levels. In cases of Xarelto overdose with bleeding, immediate cessation of the medication is vital, followed by appropriate interventions to control the bleeding. [2,3]. If the overdose occurred recently, administering activated charcoal may reduce further drug absorption from the gastrointestinal tract. Depending on the severity of the bleeding and available resources, reversal agents such as andexanet alfa or pro-coagulant therapies may be considered. Close observation and supportive care are essential in managing potential complications from the overdose [1,6]. The most frequent side effects of Xarelto (rivaroxaban) stem from its anticoagulant effects, which increase the likelihood of bleeding. These effects include bleeding incidents, limb discomfort, itching (pruritus), and muscle pain (myalgia) [1,5,7].

The liver is a central organ in maintaining proper blood clotting, as it produces most of the clotting factors and their regulatory inhibitors involved in the coagulation process [8]. When liver function is impaired, its capacity to synthesize these components is diminished, increasing the likelihood of bleeding due to impaired hemostasis. The extent of liver dysfunction is often evaluated using the Child–Pugh classification system, where Class A denotes mild impairment, Class B moderate, and Class C severe liver damage [9]. Studies have shown a direct relationship between the decline in clotting factor levels and the severity of liver impairment, as indicated by the Child–Pugh score. Additionally, liver dysfunction can significantly impact the metabolism of drugs processed by the liver, potentially leading to elevated systemic drug levels and necessitating careful dose adjustments [8,9].

Various regulatory agencies, including the International Conference on Harmonisation (ICH), the U.S. Food and Drug Administration (FDA), and the European Directorate for the Quality of Medicines and HealthCare, stress the importance of accurately estimating drugs and confirming whether the method employed is stability-indicating [10,11]. While several methods have been documented in the literature for the estimation of rivaroxaban in active pharmaceutical ingredients (APIs), including patents and journal publications, few studies incorporate force degradation data. Several analytical techniques have been described for quantifying rivaroxaban in active pharmaceutical ingredients (APIs) and dosage forms using high-performance liquid chromatography (HPLC) [4]. Typically, rivaroxaban is detected with a retention time of around 3.5 min; however, such early elution can hinder the accurate identification of impurities and degradation products. Furthermore, employing a highly restricted linear concentration range (0.005 to 40.0 µg/mL) in forced degradation studies presents significant obstacles to developing a dependable analytical method. Such a narrow range may limit the method’s sensitivity and accuracy, reducing its effectiveness in detecting degradation products and assessing the drug’s stability under stress conditions. This limited range may compromise the accuracy of results and fails to provide a comprehensive assessment of the drug’s stability profile or degradation behavior [4]. However, a comprehensive review of the literature revealed no published data on the quantitative evaluation or structural characterization of this impurity. This highlights the critical need for a more robust and reliable stability-indicating analytical method. Although certain approaches exist, a fully validated method that incorporates comprehensive stress testing has yet to be clearly defined. Such methods are vital for ensuring the accuracy and reliability of stability data, particularly for Drug Master File (DMF) submissions, and are mandatory to meet the standards set by regulatory authorities. Therefore, Given the novelty of this impurity, the aim of this study is to develop and validate a stability-indicating analytical method for the estimation of rivaroxaban impurities profile, in accordance with ICH guidelines.

## 2. Results and Discussion

Rivaroxaban is an orally administered anticoagulant that directly inhibits Factor Xa, thereby blocking the conversion of prothrombin to thrombin and, ultimately, preventing the formation of blood clots. It is commonly prescribed to reduce the risk of stroke in patients with non-valvular atrial fibrillation and to prevent deep vein thrombosis (DVT) following hip or knee replacement surgery. Several reverse-phase high-performance liquid chromatography (RP-HPLC) methods have been developed for the quantification of rivaroxaban in pharmaceutical formulations. One such method utilized a Phenomenex Luna C18 column (250 × 4.6 mm, 5 µm) maintained at 40 °C, with a mobile phase consisting of acetonitrile and water (55:45 *v*/*v*). The analysis was performed using UV detection at 249 nm with the flow rate set at 1.2 mL/min. Rivaroxaban eluted at 3.37 min, and the method was validated in accordance with ICH guidelines [10]. Another study employed a HiQSil C18 column (250 × 4.6 mm, 5 µm) at room temperature, using a mobile phase of methanol and water (65:35 *v*/*v*), a flow rate of 1.4 mL/min, and detection at 249 nm. The method demonstrated a retention time of 3.12 min and was successfully validated, confirming its accuracy and precision for the intended analytical application. Rivaroxaban elutes at a retention time of 3.5 min; however, this early elution may hinder the detection of degradants and impurities. Additionally, the very narrow linear concentration range (0.005 to 40.0 µg/mL) used in forced degradation studies is problematic and not recommended for method development [3]. This limitation can result in errors and does not adequately assess the stability of the drug or the presence of degradation products. Furthermore, many existing methods lack comprehensive stability-indicating analyses, highlighting the need for improved methodologies that can effectively evaluate the stability of rivaroxaban in tablet formulations, ensuring its potency, safety, and efficacy.

To assess the stability of rivaroxaban under various stress conditions, the raw material was subjected to acid, base, oxidative, thermal, and photolytic degradation studies, followed by analysis using an HPLC system. Table 1 presents the optimized chromatographic conditions established for the analysis of rivaroxaban, ensuring reliable separation and detection [4]. Acid hydrolysis was conducted using different concentrations of hydrochloric acid (6 N, 3 N, 1 N, 0.5 N, 0.1 N, 0.05 N, and 0.01 N HCl), with the degradation results summarized in Table 2. The findings indicate that rivaroxaban is unstable in acidic environments, with optimal degradation (5–10%) observed at 0.01 N HCl after 24 h at 75 °C. Figure 2 and Figure 3 display the chromatograms obtained from the acid degradation study, illustrating the stability profile of rivaroxaban under acidic conditions. Similarly, base hydrolysis was performed using varying concentrations of sodium hydroxide, as summarized in Table 3, demonstrating that rivaroxaban is highly susceptible to degradation under basic conditions. The optimal degradation (5–10%) was achieved with 0.01 N NaOH after one hour at 75 °C, and the corresponding chromatograms are shown in Figure 4 and Figure 5. Oxidative stress studies were conducted using hydrogen peroxide at different concentrations, and the results, presented in Table 4, confirm that rivaroxaban is unstable under oxidative conditions, with optimal degradation occurring at 0.05% H_2_O_2_ after 24 h at 75 °C. Chromatograms from the oxidative degradation study are provided in Figure 6 and Figure 7. In contrast, thermal degradation studies, in which rivaroxaban was exposed to 75 °C for 24 h, did not result in significant degradation compared to hydrolytic and oxidative conditions, as shown in Table 5 and Figure 8. Similarly, photolytic degradation studies, conducted under continuous light exposure for 24 h, showed no significant degradation compared to other stress conditions, as presented in Table 6 and Figure 9. These findings highlight the stability profile of rivaroxaban under different stress conditions and underscore the importance of a robust stability-indicating analytical method. Additionally, three different baseline control experiments using only the solvents employed in the forced degradation studies 0.01 N HCl, 0.01 N NaOH, and 0.05% H_2_O_2_ without rivaroxaban. These blank samples were subjected to the same stress conditions and analytical procedures. The purpose was to identify and confirm whether any peaks originated from the solvents or reagents themselves.

Experimental conditions: Isocratic mode with a mobile phase consisting of 30:70 (*v*/*v*) acetonitrile and 25 mM potassium phosphate monobasic buffer (pH 2.9); a flow rate set at 1.0 mL/min; detection performed at 249 nm using a 15 µL injection volume; the following column specifications: Thermo Hypersil ODS C18 (4.6 × 250 mm, 5 µm); and an analysis conducted at ambient temperature.

Analytical setup: Isocratic elution using a 30:70 (*v*/*v*) mixture of acetonitrile and 25 mM potassium phosphate monobasic buffer (pH 2.9); a flow rate set at 1.0 mL/min; UV detection at 249 nm; a 15 µL injection volume; and the separation carried out on a Thermo Hypersil ODS C18 column (4.6 × 250 mm, 5 µm) under ambient conditions.

Chromatographic conditions: Isocratic elution with a mobile phase of 30% acetonitrile and 70% 25 mM potassium phosphate buffer monobasic (pH 2.9); flow rate set at 1.0 mL/min; detection wavelength at 249 nm; ambient temperature; 15 µL injection volume; and a Thermo Hypersil ODS C18 column (4.6 × 250 mm, 5 µm).

Chromatographic parameters: Isocratic elution with a mobile phase comprising 30% acetonitrile and 70% of 25 mM potassium phosphate monobasic buffer (pH 2.9); a flow rate maintained at 1.0 mL/min; detection performed at 249 nm; an injection volume of 15 µL; and the analysis conducted at ambient temperature using a Thermo Hypersil ODS C18 column (4.6 × 250 mm, 5 µm).

Chromatographic setup: Isocratic separation using a mobile phase of 30% acetonitrile and 70% 25 mM potassium phosphate monobasic buffer (pH 2.9); a flow rate set at 1.0 mL/min; detection performed at 249 nm; an injection volume of 15 µL; and the analysis conducted at room temperature on a Thermo Hypersil ODS C18 column (4.6 × 250 mm, 5 µm).

The mixed samples’ impurities from all stress conditions were injected and analyzed using the HPLC system. Rivaroxaban was detected at a retention time of 12.20 min, accompanied by eleven distinct degradation peaks. Notably, three major degradation products were observed at retention times of 8.358, 9.030, and 9.493 min, indicating significant breakdown under the applied conditions. Additionally, specific impurities were identified, including impurity G at 2.79 min, impurity D at 3.50 min; several unknown impurities at 3.20, 4.00, 4.59, and 4.77 min; impurity H at 5.32 min; impurity C at 6.14 min; impurity E at 8.36 min; impurity A at 9.03 min; and impurity F at 9.49 min, as shown in Figure 10B and Table 7. All observed peaks were distinctly separated from the rivaroxaban peak, fulfilling the specificity requirements for resolution. Additionally, a baseline control, consisting of a mixture of 0.01 N HCl, 0.01 N NaOH, and 0.05% H_2_O_2_, was also subjected to the same stress conditions and analytical procedures to further rule out any potential interference or cross-reactivity among solvents. This confirms the method’s effectiveness in accurately differentiating rivaroxaban from its associated degradation products. Furthermore, the peak purity factor surpassed the predefined threshold, confirming that the rivaroxaban peak was not affected by co-eluting substances. The specificity evaluation confirmed the method’s ability to accurately identify and quantify rivaroxaban, even when coexisting with its degradation products and possible impurities, ensuring precise and selective analysis. Supporting chromatographic data, including a peak purity plot and a 3D chromatographic profile of rivaroxaban, are provided in Appendix A, demonstrating compliance with ICH and FDA regulatory standards [10].

Chromatographic conditions: Isocratic mode with a mobile phase consisting of 30% acetonitrile and 70% of 25 mM potassium phosphate monobasic buffer (adjusted to pH 2.9), flow rate maintained at 1.0 mL/min, detection at 249 nm, injection volume of 15 µL, analysis conducted at ambient temperature using a Thermo Hypersil ODS C18 column (4.6 × 250 mm, 5 µm).

The system suitability test results, as detailed in Table 8, demonstrate the robustness and consistency of the developed analytical method. All chromatographic peaks showed excellent resolution, with tailing factors approaching one, indicating well-shaped and symmetrical peaks. Column efficiency was high, as evidenced by the number of theoretical plates exceeding 9000, reflecting effective separation performance. For working standard solution #1, the relative standard deviations (%RSDs) for the peak area and retention time across six replicates were 0.77% and 0.86%, respectively. For working standard solution #2, two replicate injections yielded %RSD values of 0.32% and 0.24%. These results, as illustrated in Figure 11, confirm the method’s precision. Additionally, the percent drift remained below 1%, supporting the method’s stability and reproducibility. All parameters met the acceptance criteria outlined in ICH guidelines, affirming the method’s suitability for subsequent analytical applications.

Chromatographic conditions: The separation was carried out using an isocratic elution with a mobile phase composed of 30% acetonitrile and 70% 25 mM potassium phosphate monobasic buffer (pH adjusted to 2.9). The flow rate was set to 1.0 mL/min, with detection performed at 249 nm. A 15 µL volume was injected at ambient temperature using a Thermo Hypersil ODS C18 analytical column (4.6 × 250 mm, 5 µm particle size).

To further evaluate the robustness of the analytical method, alternative mobile phases were prepared by adjusting the buffers’ pHs to 3.1 and 2.7, in accordance with the established protocol. Each modified mobile phase was tested separately on the HPLC system to observe its effect on retention time and peak morphology. This comparative analysis enabled a thorough assessment of the method’s performance under minor pH fluctuations, confirming its reliability and aiding in the optimization of chromatographic separation conditions. Robustness testing also included variations in the flow rate (1.2, 1.0, and 0.8 mL/min), detection wavelength (247, 249, and 251 nm), % B composition (25%, 30%, and 35%), and injection volume (13, 15, and 17 μL). The results confirmed that all tested parameters met the acceptance criteria, demonstrating the method’s reliability under deliberate changes and ensuring its suitability for routine analysis. The findings from the robustness study, as outlined in Appendix A, confirmed the method’s resilience under slight variations in the analytical conditions, in accordance with the ICH’s guidelines. Tailing factors remained within the acceptable range of 0.9 to 2, and theoretical plate counts consistently exceeded 2000, indicating sufficient column efficiency. The method proved robust against minor changes in solvent strength, flow rate, buffer pH, detection wavelength, and injection volume [4]. Supporting chromatograms and corresponding chromatographic conditions are provided in Appendix A, further validating the method’s stability and reliability.

To evaluate the linearity of rivaroxaban impurities and degradants, standard solutions at concentrations of 2 ppm, 1.75 ppm, 1.5 ppm, 1.25 ppm, and 1.0 ppm were prepared and analyzed using the HPLC system. Each surrogate sample was injected, and the corresponding peak areas were recorded as shown in Figure 12. The data presented in Table 9 were used to construct a calibration curve by plotting peak area against analyte concentration. The resulting linear regression equation for rivaroxaban’s active pharmaceutical ingredient produced a correlation coefficient (R^2^) of 0.9987, as shown in Figure 13. The strong correlation observed meets the established linearity acceptance criteria, confirming a direct and proportional relationship between analyte concentration and detector response. These results confirm the method’s reliability and appropriateness for precise quantification of impurities and degradation products over the evaluated concentration range.

Chromatographic Settings: An analysis was performed under isocratic conditions using a mobile phase consisting of acetonitrile and 25 mM potassium phosphate monobasic buffer (pH adjusted to 2.9) at a 30:70 (*v*/*v*) ratio. The flow rate was maintained at 1.0 mL/min, with detection carried out at 249 nm. A 15 µL injection volume was used, and separations were achieved on a Thermo Hypersil ODS C18 column (4.6 × 250 mm, 5 µm particle size) operated at room temperature.

Each surrogate sample prepared at concentrations of 2.0 ppm, 1.5 ppm, and 1.0 ppm was analyzed in triplicate using the HPLC system to assess the accuracy of the analytical method. The resulting peak areas, as presented in Figure 14, were used to calculate the percent recovery for each concentration level. As detailed in Table 10, recovery rates for impurities and degradants consistently fell within the acceptable range of 95% to 105% of their respective theoretical values. These results confirm that the method provides accurate and reliable quantification of the active compound, as well as associated impurities and degradation products, across the evaluated concentration range. All tested levels complied with predefined acceptance criteria, supporting the method’s robustness and its suitability for routine quality-control applications.

Chromatographic conditions: Isocratic elution, mobile phase 30:70 ACN/25 mM potassium phosphate buffer monobasic pH 2.9, flow rate 1.0 mL/min, detection wavelength at 249 nm, ambient temperature, 15 µL injection volume, Thermo Hypersil ODS C_18_ (4.6 × 250 mm, 5 µm) column.

Method precision for the determination of impurities and degradants was evaluated during the accuracy study using rivaroxaban as a surrogate. Six replicates of a 1.5 ppm rivaroxaban solutions were prepared and analyzed using the HPLC system under optimized chromatographic conditions. Peak areas corresponding to each injection were recorded, and the method’s precision was assessed by calculating the percent relative standard deviation (%RSD). As shown in Table 11, the %RSD was determined to be 1.29, which falls within the predefined acceptance limits, thereby confirming the method’s precision and reproducibility. For injection precision, a single 1.5 ppm rivaroxaban sample was prepared from a 100 ppm stock solution and injected six times into the HPLC system under the same optimized conditions. The corresponding peak areas were recorded and are illustrated in Figure 15. The percent relative standard deviation (%RSD) for injection precision was calculated to be 1.46, as detailed in Table 12, which falls within the acceptable limits for injection precision. To evaluate intermediate precision, six separate 1.5 ppm rivaroxaban samples were analyzed on a different HPLC system under the same optimized conditions (Figure 16). The %RSD values were calculated for both peak areas and retention times, yielding results of 0.38 and 0.28, respectively—each meeting the established criteria for intermediate precision, as summarized in Table 12. The chromatographic profiles and analytical conditions are presented in Figure 17. These results demonstrate the method’s robustness and reproducibility across different instruments and operating conditions.

Chromatographic Conditions: Separation was carried out using an isocratic elution with a mobile phase composed of 30% acetonitrile and 70% 25 mM potassium phosphate monobasic buffer (pH adjusted to 2.9). The analysis was performed at room temperature with a flow rate of 1.0 mL/min. Detection was conducted at a wavelength of 249 nm using a 15 µL injection volume on a Thermo Hypersil ODS C18 column (dimensions: 4.6 × 250 mm, 5 µm particle size).

Chromatographic Conditions: Analysis was conducted under isocratic conditions using a mobile phase consisting of 30% acetonitrile and 70% 25 mM potassium phosphate monobasic buffer adjusted to pH 2.9. The separation was performed on a Waters (Milford, MA, USA) XTerra RP-18 column (4.6 × 250 mm, 5 µm) at room temperature. The mobile phase was delivered at a flow rate of 1.0 mL/min, with a 15 µL injection volume. Detection was carried out at 249 nm using a UV detector.

The limit of detection (LOD) for rivaroxaban was determined by preparing a series of dilutions from a 5000 ppm stock solution, with concentrations ranging from 10.0 ppm down to 0.2 ppm. Each dilution was analyzed using the HPLC system, and the resulting chromatograms (Appendix A) were assessed to calculate the signal-to-noise (S/N) ratios. As shown in Appendix A, an S/N ratio of 4.0 was observed at 0.3 ppm. Since this exceeds the minimum criterion of S/N ≥ 3, 0.3 ppm was established as the LOD. To determine the limit of quantitation (LOQ), a 1.0 ppm rivaroxaban solution was injected into the HPLC system ten times. The resulting peak areas were recorded, and their signal-to-noise ratios and the relative standard deviation (%RSD) were calculated. The %RSD was found to be 1.54%, well within the acceptable threshold of ≤10% for LOQ determination. This confirms that 1.0 ppm meets the criteria for accurate and reliable quantification. Additionally, in alignment with International Council for Harmonisation (ICH) guidelines, the LOD and LOQ values were supported by calculations using the standard deviation of the response and the slope of the calibration curve, further validating the method’s sensitivity [10]. The formulas for calculating LOD and LOQ are as follows:$$LOD=3.3\times (\frac{\sigma }{S})\;\mathrm{and}\;LOQ=10\times (\frac{\sigma }{S})$$
where σ is the standard deviation of the response, and S is the slope of the calibration curve.

From the linearity assessment, the standard deviation of the response (σ) was calculated as 96.07, and the slope of the calibration curve (S) was 971.04. Based on these values, the limit of detection (LOD) and limit of quantitation (LOQ) were determined using the ICH-recommended formulas as follows: LOD = 3.3 × (σ/S) and LOQ = 10 × (σ/S). The resulting values were 0.33 ppm for LOD and 0.99 ppm for LOQ, which closely corresponds to the signal-to-noise ratio-based determinations (LOD = 0.31 ppm; LOQ = 1.0 ppm). These findings are in accordance with both the ICH’s and USP’s guidelines for analytical method validation, reinforcing the sensitivity, accuracy, and robustness of the developed method [10,11].

## 3. Materials and Methodology

### 3.1. Chemicals and Reagents [4]

Hydrochloric acid 12 N, Fisher Scientific, Hampton, NH, USA, CAS A144-212, lot no. 135238; acetonitrile (ACN) HPLC grade, Fisher Scientific, CAS 7778-77; sodium hydroxide, EM Science, Hatfield, PA, USA, lot no. 33257-344; hydrogen peroxide 30%, Fisher Scientific, CAS 7722-84-1, lot no. 094115; potassium phosphate monobasic, Merck & Co., Boston, MA, USA; CAS7758-11-4, lot no. A0340250; phosphoric acid (85%) J.T Baker, Phillipsburg, NJ, USA, lot no. 33397; deionized Water (in house system); pH buffer standards: 4.0, Fischer Scientific, lot no. 122994; pH buffer standards: 7.0, Fischer Scientific, lot no. 127454; pH buffer standards: 9.0, Fischer Scientific, lot no. 123465; Rivaroxaban USP, CAS: 366789-02-8, lot no. F10350; Rockville, MD, USA.

### 3.2. Chromatography Instrument

The analysis was carried out using an Agilent high-performance liquid chromatography (HPLC) system, consisting of the following components [4]:Agilent 1100 Series HPLC System with DAD Detector (Santa Clara, CA, USA), equipped with:➢G1322A Degasser, serial no. JP05033159;➢G1311A Quaternary Pump, serial no. DE111155886;➢G1316A Column Thermostat, serial no. DE11123026;➢G1329A ALS (Thermostat Auto sampler), serial no. DE11115876;➢G1365B DAD (Diode Array Detector), serial no. DE11101219.Data Acquisition System (ChemStation) for HPLC (Agilent Technologies, Santa Clara, CA, USA);C18 (4.6 × 250 mm, 5 µm) Thermo ODS Hypersil, Part No; 30105-254630, serial no. 0153571S, Waltham, MA, USA.

The equipment was operated under carefully optimized chromatographic conditions. All components were calibrated and maintained in accordance with the manufacturers’ guidelines to ensure consistent performance and accurate data acquisition.

### 3.3. Additional Equipment

The following additional equipment was utilized during the analysis:○Sonicator/Ultrasonic: T21; Danbury, CT, USA○Analytical balance: Mettler Toledo, Model xp205, serial no. 1129211544; Greifensee, Switzerland.○pH meter: Accumet Research, Fisher Scientific, serial no. AR93311993; Hampton, NH, USA○pH (0–13.0 stripes): J.T Baker Inc., 4403-01; Center Valley, PA, USA○UV light: Spectroline, Model ENF-260C, serial no. 1358236; Westbury, NY, USA○Hot plate/Stirrer: Corning, Corning, NY, USA, Model PC-420, serial no. 420505018647○Filter: 0.45 µm PVDF (polyvinylidene difluoride) membrane; Corning, NY, USA○Disposable syringe filter: Nylon Acordis CR 25 mm 0.2 µm, HPLC certified, Corning, NY, USA

### 3.4. Chromatographic Conditions

The optimized chromatographic parameters for the analysis of rivaroxaban are outlined in Table 1 [4]. These parameters were precisely optimized to maintain accurate analyte separation, support precise quantification, and reduce overall analysis time with harmonious and dependable discovery.

### 3.5. Solution Preparation Procedures

The detailed procedures for the preparation of solutions used across all validation parameters, including linearity, mobile phase composition, solution stability, limit of quantitation (LOQ), limit of detection (LOD), specificity, robustness, system suitability, accuracy, and precision, are provided in the Appendix A. These protocols were followed consistently to ensure the reliability and reproducibility of the analytical method throughout the validation process [4].

### 3.6. Stock Solution of Rivaroxaban (5000 ppm)

Accurately weigh 250 mg of rivaroxaban and transfer it into a 50 mL volumetric flask. Add approximately 25 mL of a solvent mixture consisting of acetonitrile and deionized water in a 70:30 (*v*/*v*) ratio. Sonicate the mixture for 20 min or until the drug is fully dissolved. Once dissolution is complete, dilute to volume with the same solvent mixture (ACN: DI water, 70:30 *v*/*v*) and mix thoroughly to ensure homogeneity.

### 3.7. Stock Solution of Rivaroxaban (3500 ppm)

Precisely weigh 350 mg of rivaroxaban and transfer it into a 100 mL volumetric flask. Add approximately 25 mL of a solvent mixture composed of acetonitrile and deionized water in a 70:30 (*v*/*v*) ratio. Sonicate the solution for 20 min or until the drug is fully dissolved. After complete dissolution, dilute to the final volume with the same ACN: DI water (70:30 *v*/*v*) mixture. Cap and shake the flask thoroughly to ensure uniform distribution.

### 3.8. Mobile Phase B (100% Acetonitrile)

Measure and transfer 1000 mL of acetonitrile (ACN) into a clean mobile phase reservoir. Sonicate the solvent for 20 min to effectively degas the solution and eliminate any dissolved air bubbles that may interfere with chromatographic performance.

### 3.9. Mobile Phase A (Buffer pH 2.9): To Prepare One Liter of 25 mM Potassium Phosphate Monobasic Solution at pH 2.90

Accurately weigh 3.40 g of potassium phosphate monobasic and transfer it into a 1000 mL beaker. Add 1000 mL of deionized (DI) water and stir continuously until the salt is completely dissolved. Carefully adjust the pH to 2.9 (using calibrated pH meter) by adding phosphoric acid dropwise while maintaining constant stirring. The solution was filter through a 0.45 µm membrane filter. Finally, the filtered buffer was sonicated for 20 min to eliminate any air bubbles or dissolved gases prior to use.

### 3.10. Method Development and Optimization

In the development of the HPLC analytical methods, evaluating the polarity characteristics, namely, the hydrophilicity or hydrophobicity, of the analyte is essential for choosing suitable stationary and mobile phases [12]. For lipophilic compounds, such as rivaroxaban, reverse-phase high-performance liquid chromatography (RP-HPLC) is generally the method of choice. This approach employs a hydrophobic stationary phase alongside a polar mobile phase, effectively promoting retention and resolution of nonpolar molecules [13]. This study aimed to design a robust, straightforward, and highly selective isocratic RP-HPLC method tailored for the quantification of impurities in raw rivaroxaban material. The approach prioritized the direct and efficient resolution of the parent compound from its associated degradation products and impurities, eliminating the need for complex sample pretreatment. Method optimization was carried out by systematically adjusting the chromatographic parameters to achieve the optimal column performance, which was reflected in the high theoretical plate counts, sharp and symmetrical peaks, low tailing factors, and strong separation efficiency. The method, originally refined and validated through earlier investigations conducted by our team, has been adapted here to meet the specific analytical demands of this study [4]. To establish optimal chromatographic conditions and ensure methodological precision, several key assessments were undertaken, including solubility profiling, infrared (IR) spectral analysis, selection of detection wavelength, column type, pH adjustment, injection volume optimization, isocratic elution evaluation, nominal concentration determination, and stress degradation studies. For all subsequent preparations, a 70:30 *v*/*v* acetonitrile (ACN) to deionized water solution was selected as the diluent. The detection wavelength was set at 249 nm, and a strong linear correlation was observed across the concentration range of 50 to 1000 ppm. A 15 µL injection volume and a working concentration of 700 ppm were established as optimal. Chromatographic separation was achieved using a Thermo Hypersil ODS C18 column (4.6 × 250 mm, 5 µm particle size). The most effective mobile phase was determined to be a 70:30 *v*/*v* mixture of phosphate buffer (pH 2.9) and ACN [4].

### 3.11. Forced Degradation Study

The stability of rivaroxaban, as a raw material, is a critical determinant of its overall potency, purity, and safety. Instability in the compound can lead to the generation of harmful degradation products or a diminished therapeutic effect due to reduced active content. Therefore, it is essential to evaluate its purity profile under various environmental conditions [13]. Forced degradation studies, also referred to as stress studies, were conducted following the ICH’s guidelines. According to the acceptance criteria, the optimal degradation range for stability studies should be between 5 and 10%. In these experiments, rivaroxaban was tested at a final concentration of 700 ppm under the following stress conditions [10]:Acid hydrolysis;Base hydrolysis;Oxidation with hydrogen peroxide;Thermal degradation (heat);UV light exposure.

The percentage degradation of the forced degradation sample was determined using the following equation.$$\%\;Degradation=\frac{A_{sample}{-A}_{control}}{A_{control}}\times 100$$
where A_control_ is the peak area of the control sample, and A_sample_ is the peak area of the degraded sample.

### 3.12. Solution Preparation for Forced Degradation Study

#### 3.12.1. Stock Solution of Rivaroxaban (3500 ppm)

Accurately weigh 350 mg of rivaroxaban and transfer it into a 100 mL volumetric flask. Add 25 mL of ACN: H_2_O at a 70:30 (*v*/*v*) ratio into a 100 mL flask. Sonicate the solution for approximately 20 min or until complete dissolution is achieved. After ensuring full dissolution, make up the volume to the 100 mL mark with the same ACN:DI water (70:30 *v*/*v*) mixture, then mix thoroughly to ensure uniformity.

#### 3.12.2. Preparation of 6 N HCl Stock Solution

Transfer 49.6 mL of concentrated hydrochloric acid (12.1 N) into a 100 mL volumetric flask. Dilute to the mark with deionized (DI) water and mix thoroughly.

#### 3.12.3. Preparation of Serial Dilutions of HCl

Using the 6 N HCl stock solution, prepare the following dilutions, following the procedure detailed in the Appendix A: 3 N HCl, 1 N HCl, 0.5 N HCl, 0.1 N HCl, 0.05 N HCl, and 0.01 N HCl.

#### 3.12.4. Preparation of 6 N NaOH Stock Solution

Weigh 24 g of sodium hydroxide and dissolve it in 75 mL of deionized (DI) water. Transfer the solution into a 100 mL volumetric flask, dilute to the mark with DI water, and mix thoroughly.

#### 3.12.5. Preparation of Serial Dilutions of NaOH

Using the 6 N NaOH stock solution, prepare the following dilutions: 3 N NaOH, 1 N NaOH, 0.5 N NaOH, 0.1 N NaOH, 0.05 N NaOH, and 0.01 N NaOH, following the procedure outlined in the Appendix A.

#### 3.12.6. Preparation of 3% H_2_O_2_ Stock Solution

Transfer 10 mL of 30% H_2_O_2_ solution into a 100 mL volumetric flask. Add DI water to complete the volume to 100 mL and shake it thoroughly.

#### 3.12.7. Preparation of Serial Dilutions of H_2_O_2_

Using the 3% H_2_O_2_ stock solution, prepare the following dilutions, following the procedure outlined in the Appendix A: 1% H_2_O_2_, 0.5% H_2_O_2_, 0.1% H_2_O_2_, and 0.05% H_2_O_2_.

### 3.13. Acid Degradation

To assess the acid degradation, a control sample (700 ppm) was prepared from the rivaroxaban stock solution (3500 ppm) and analyzed using the HPLC system to determine the percentage degradation of the stressed sample. A chromatogram of the control sample is shown in Figure 2. Acid stress samples were prepared using 6 N HCl, 3 N HCl, 1 N HCl, 0.5 N HCl, 0.1 N HCl, 0.05 N HCl, and 0.01 N HCl, as detailed in the Appendix A. Below is the procedure for preparing an acid stress sample with 0.01 N HCl. All prepared acid-stressed samples were subsequently injected into the HPLC system, and chromatograms were recorded to assess the extent of rivaroxaban degradation under acidic conditions.

#### Acid Stress Sample Preparation Degraded with (0.01 N HCl)

Transfer 2 mL of the rivaroxaban stock solution (3500 ppm) into a screw-cap test tube; then, add 2 mL of 0.01 N HCl. Heat the mixture on a heating block at 75 °C for 24 h. After heating, allow the solution to cool to room temperature; then, add 2 mL of 0.01 N NaOH to neutralize the acid. Carefully transfer the neutralized solution into a 10 mL volumetric flask and dilute to the mark with a 30:70 (*v*/*v*) mixture of DI water and ACN. Shake thoroughly to obtain a final concentration of 700 ppm. Before injecting the solution into the HPLC system, verify that the pH is neutral (pH 7) using pH strips. Finally, filter the solution through a 0.45 µm membrane filter before injection into the HPLC system.

### 3.14. Base (Alkali) Degradation

To assess the degradation under basic conditions, a test sample with a concentration of 700 ppm was prepared by diluting the primary rivaroxaban stock solution (3500 ppm). This sample was then subjected to HPLC analysis to quantify the extent of degradation and calculate the percentage loss due to alkaline stress. The control sample chromatogram is illustrated in Figure 4. Base stress samples were prepared using 6 N NaOH, 3 N NaOH, 1 N NaOH, 0.5 N NaOH, 0.1 N NaOH, 0.05 N NaOH, and 0.01 N NaOH, as detailed in the Appendix A. Below is the procedure for preparing a base stress sample with 0.01 N NaOH. Each of the prepared samples was introduced into the HPLC system, and chromatographic profiles from the base-induced degradation study of rivaroxaban were captured and documented for analysis.

#### Base Stress Sample Preparation Degraded with (0.01 N NaOH)

Transfer 2 mL of rivaroxaban stock solution (3500 ppm) into a screw-cap test tube; then, add 2 mL of 0.01 N NaOH. Heat the mixture on a heating block at 75 °C for 24 h. After cooling to room temperature, add 2 mL of 0.01 N HCl to neutralize the basic solution. Accurately transfer the neutralized solution into a 10 mL volumetric flask; then, dilute to the mark with a 30:70 (*v*/*v*) DI water: ACN mixture and shake thoroughly to obtain a final concentration of 700 ppm. Before injecting the sample into the HPLC system, it is crucial to check the pH using pH strips to ensure neutrality (pH 7). Finally, filter the sample through a membrane filter (0.45 µm) before injection into the HPLC system.

### 3.15. Hydrogen Peroxide Degradation (Oxidation)

To evaluate oxidative degradation, a control sample with a concentration of 700 ppm was prepared by diluting the rivaroxaban stock solution (3500 ppm) and analyzed using HPLC to calculate the percentage degradation under oxidative stress. The chromatogram of this control sample is presented in Figure 6. Oxidative stress conditions were studied using hydrogen peroxide at concentrations of 30%, 3%, 1%, 0.5%, 0.1%, and 0.01%, as outlined in the Appendix A. The following procedure was used for preparing the 0.01% H_2_O_2_ stress sample: An appropriate volume of the stock solution was mixed with 0.01% hydrogen peroxide and allowed to react for a specified duration. After the reaction period, the solution was diluted with the appropriate diluent and then injected into the HPLC system. The chromatograms obtained from these oxidative stress samples were recorded to evaluate the degradation behavior of rivaroxaban under varying oxidative conditions.

#### Oxidation Stress Sample Preparation Degraded with (0.05% H_2_O_2_)

Transfer 2 mL of rivaroxaban stock solution (3500 ppm) into a screw-cap test tube, add 2 mL of 0.05% H_2_O_2_, and heat the mixture on a heating block at 75 °C for 24 h. After cooling to room temperature, transfer the solution into a 10 mL volumetric flask, dilute to the mark with a 30:70 (*v*/*v*) mixture of DI water and ACN, and mix thoroughly to achieve 700 ppm a final concentration. Moreover, filter the solution before injection into the HPLC system using a 0.45 µm membrane filter.

### 3.16. Thermal Degradation (Heat)

For the thermal degradation study, a control sample with a concentration of 700 ppm was prepared by diluting the 3500 ppm rivaroxaban stock solution. This control was then analyzed using the HPLC system to determine the percentage degradation of the heat-exposed sample. The chromatographic profile of the control sample is presented in Figure 8A.

#### Solution Preparation for Thermal Degradation

Transfer 2 mL of the rivaroxaban stock solution (3500 ppm) into a screw-cap test tube and heat it in a heating block at 75 °C for 24 h. After heating, allow the solution to cool to room temperature. Transfer the entire solution into a 10 mL volumetric flask and complete to the mark with a mixture of DI water and ACN (30:70 *v*/*v*). Shake the flask thoroughly to achieve a final concentration of 700 ppm. Filter the solution using a 0.45 µm membrane filter before injecting it into the HPLC system. The chromatograms of the rivaroxaban sample are shown in Figure 8B.

### 3.17. Photolysis (UV Light) Stress Study

To evaluate the photolytic degradation, a 700 ppm control sample was obtained by diluting a 3500 ppm rivaroxaban stock solution. This control was analyzed using the HPLC system to determine the extent of degradation following light exposure. The resulting chromatographic profile of the control sample is presented in Figure 9A.

#### Solution Preparation for Photolysis

To perform the UV-induced degradation study, place approximately 50 mg of rivaroxaban raw material into a cuvette, and expose it to ultraviolet light for a duration of 24 h. After exposure, accurately transfer 17.50 mg into a 25 mL volumetric flask. To this, add 15 mL of a solvent mixture containing acetonitrile and deionized water (70:30 *v*/*v*), followed by sonication for 20 min or until complete dissolution of the drug. Bring the solution to a certain volume with the same solvent mixture and mix thoroughly to achieve a final concentration of 700 ppm. Prior to the HPLC injection, filter the sample using a 0.45 µm membrane filter. The resulting chromatogram from the UV degradation study of rivaroxaban is displayed in Figure 9B.

### 3.18. Mixed Degradation Study (Acid, Base, and Oxidation)

The purpose of the mixed degradation study was to assess the efficiency of the developed HPLC method in distinctly separating rivaroxaban from all possible degradation products and impurities, ensuring precise quantification and purity evaluation [10,14]. A 700 ppm control sample was prepared to establish a reference point by diluting a 3500 ppm rivaroxaban stock solution; then, it was analyzed via HPLC. The resulting chromatogram, as illustrated in Figure 10A, served as the baseline for determining the extent of degradation under combined stress conditions. This analysis demonstrated the method’s ability to effectively differentiate rivaroxaban from its degradation by-products. For the mixed degradation test, 1 mL from each mildly degraded sample 0.01 N HCl (24 h), 0.01 N NaOH (1 h), and 0.05% H_2_O_2_ (24 h) was combined in a test tube. The mixture was thoroughly mixed, passed through a 0.45 µm membrane filter, and injected into the HPLC system. The chromatogram obtained from this combined stress sample is presented in Figure 10B. Additionally, a control solution consisting of 0.01 N hydrochloric acid, 0.01 N sodium hydroxide, and 0.05% hydrogen peroxide was also subjected to the same stress conditions to rule out the possibility of any impurities originating from the solvents used.

The investigation identified that rivaroxaban’s major degradation products appeared at retention times of 2.79, 3.20, 3.50, 4.00, 4.59, 4.77, 5.32, 6.14, 8.36, 9.03, and 9.49 min. Each of these peaks was distinctly separated from the rivaroxaban peak, which eluted at around 12 min. This complete resolution highlights the effectiveness of the developed HPLC method in clearly distinguishing the active drug from its breakdown products. The successful separation confirms the method’s robustness and accuracy in both identifying and quantifying rivaroxaban, with minimal interference from degradation compounds. A comprehensive overview of degradation patterns across various stress conditions is included in Appendix A.

### 3.19. Method Validation

To guarantee compliance with Good Laboratory Practice (GLP) and Good Manufacturing Practice (GMP), the newly developed analytical method underwent rigorous validation in line with major international regulatory frameworks, including the ICH’s guidelines (Q2A and Q2B), the U.S. FDA, and the United States Pharmacopeia (USP) [10,11,15]. The validation process encompassed a comprehensive evaluation of key analytical parameters to establish the method’s precision, reliability, and robustness, as follows:

**System Suitability Testing (SST);** Confirmed consistent system performance before sample analysis to ensure valid results.

**Specificity;** demonstrated the method’s capability to accurately identify rivaroxaban in the presence of related impurities and degradation products, without interference.

**Robustness;** tested the method’s resilience to slight changes in analytical conditions, such as pH variation or alterations in the mobile phase composition.

**Solution Stability;** assessed the chemical stability of rivaroxaban in solution over the analytical time frame, ensuring accurate readings without degradation.

**Linearity and Analytical Range;** verified that the method produces a direct, proportional response across a defined concentration span.

**Accuracy and Precision;** method precision—evaluated reproducibility within the method across multiple replicates.

**Injection Precision;** assessed consistency in peak areas across multiple injections.

Intermediate Precision: measured variability between different analysts and test days.

**Limits of Detection (LOD) and Quantitation (LOQ);** Identified the smallest concentrations of impurities and degradants that can be quantified and reliably detected.

Successful validation against these criteria confirms that the method is suitable for high-quality routine analysis of rivaroxaban, offering consistent performance in accordance with global regulatory expectations.

### 3.20. System Suitability Study for Impurities and Degradants

System suitability testing (SST) is a fundamental prerequisite for ensuring the reliability of high-performance liquid chromatography (HPLC) results, especially in the analysis of pharmaceutical impurities and degradation products. In accordance with the ICH’s Q2(R1) guidelines, SST serves to confirm that all components of the chromatographic system, including instrumentation, software, reagents, and analytical conditions, are functioning optimally prior to sample analysis [10]. The purpose of SST is to validate the system’s readiness and guarantee that it can deliver accurate and reproducible results throughout the testing process. This is particularly vital in impurity profiling, where minor variations can significantly affect the detection and quantification of degradation products. The acceptance limits for key SST parameters, as recommended by international regulatory authorities, include the following.

% Relative standard deviation (RSD) of the peak response (n ≥ 6): not more than 1.0%;% RSD of retention times (n ≥ 6): not more than 1.0%;Theoretical plate count (column efficiency): minimum of 2000 plates;Tailing factor: within the range of 0.9 to 2.0;Capacity factor (k’): at least 2.0;Resolution between critical peaks: minimum of 2.0;Percentage drift in peak area or retention time: maximum 2%.

By satisfying these stringent criteria, the HPLC system is verified to be robust and consistent, ensuring that the analytical method performs within validated specifications. This process underpins the integrity of the impurity and degradant assessments and supports compliance with global quality and regulatory standards.

### 3.21. Standard Solutions Preparation for System Suitability

#### 3.21.1. Stock Solution of Rivaroxaban (10 ppm)

To prepare the initial diluted rivaroxaban solution, pipette 0.1 mL from the 5000 ppm stock solution into a 50 mL volumetric flask. Fill the flask to volume with a solvent mixture of acetonitrile and deionized water in a 70:30 (*v*/*v*) ratio, and mix thoroughly to ensure uniformity.

#### 3.21.2. Working Standard Solution of Rivaroxaban (1.5 ppm) #1

Measure 3.75 mL from a 10 ppm intermediate rivaroxaban solution and transfer it into a 25 mL volumetric flask. Dilute to the mark using the same ACN:DI water (70:30 *v*/*v*) solvent system and mix thoroughly.

#### 3.21.3. Working Standard Solution of Rivaroxaban (1.5 ppm) #2

Measure 3.75 mL from a 10 ppm intermediate rivaroxaban solution and transfer it into a 25 mL volumetric flask. Dilute to the mark using the same ACN:DI water (70:30 *v*/*v*) solvent system and mix thoroughly.

To evaluate the system’s performance, both working standard solutions—identified as Solution #1 and Solution #2—were subjected to HPLC analysis. Solution #1 was injected into the HPLC system six times consecutively to test repeatability, while Solution #2 was injected twice to assess short-term consistency. The resulting chromatograms, presented in Figure 11, were analyzed to determine the key system suitability parameters. Calculated metrics included the relative standard deviation (%RSD) of both peak area and retention time for each set of injections. These values confirmed the consistency of the HPLC system in detecting rivaroxaban. Additionally, the tailing factor and the number of theoretical plates were recorded for each peak to evaluate peak symmetry and column efficiency. Percent drift (%Drift) was also computed using the following formula:$$\%\;Drift=\frac{X1-X2}{X1}\times 100$$
where X1 is the average peak area of six replicate injection of working standard solution #1, and X2 is the average peak area of two replicate injections of working standard solution #2.

### 3.22. Specificity

Specificity is a critical validation parameter that confirms the method’s ability to selectively detect and quantify rivaroxaban in the presence of structurally related impurities, degradation products, and other sample matrix components. This characteristic ensures that the rivaroxaban peak is clearly distinguished from all other substances present in the chromatographic run, eliminating any risk of analytical interference. To meet regulatory standards, the following acceptance criteria were established for specificity [10]:▪A peak purity index exceeding 990, indicating a single, uncontaminated peak with no overlapping signals;▪Chromatographic separation must achieve a resolution value greater than 2 between rivaroxaban and adjacent peaks, confirming distinct baseline separation;▪The rivaroxaban peak must remain unaffected by any co-eluting impurities or degradation by-products.

This analysis is designed to validate the method’s selectivity by ensuring that rivaroxaban can be quantified accurately and independently from all other components, confirming the integrity and reliability of the measurement. For the mixed degradation study, degradation solutions of acid, base, and oxidative conditions each exhibiting less than 10% degradation were combined [4]. Specifically, 1 mL of each solution was mixed:○0.01 N HCl (8.8% degradation over 24 h);○0.01 N NaOH (8.1% degradation over one hour);○0.05% H_2_O_2_ (6.5% degradation over 24 h).

The combined sample was thoroughly mixed to ensure uniformity and subsequently passed through a 0.45 µm membrane filter. It was then analyzed using an Agilent 1100 HPLC system fitted with a diode array detector (DAD). The outcomes of this analysis were documented in a prior publication of ours.

### 3.23. Method Robustness

Robustness testing is a vital component of analytical method validation, aimed at determining the method’s stability under small, intentional variations in experimental conditions. In alignment with the ICH’s recommendations, this evaluation confirms the method’s dependability when applied under realistic laboratory scenarios [4,10]. In the present investigation, the following five key parameters were systematically altered to assess the method’s robustness:**Buffer pH**: (2.9 ± 0.2);**Flow Rate**: (1.0 ± 0.2 mL/min);**Wavelength**: (249 ± 2 nm);**% B Composition**: (30 ± 5% B);**Injection Volume**: (15 ± 2 μL).

To verify robustness, the following performance criteria were applied [10]:**Tailing factor**: Between 0.9 and 2.0;**Number of theoretical plates**: ≥2000;**Peak resolution (RS)**: >2.0, ensuring baseline separation.

### 3.24. Linearity and Range for (Rivaroxaban) Impurities and Degradants

In this investigation, rivaroxaban served as a representative compound to assess the detector’s sensitivity toward potential impurities and degradation products. A concentrated stock solution of rivaroxaban (5000 ppm) was utilized to create a range of working solutions at varying concentrations: 2.00 ppm, 1.75 ppm, 1.50 ppm, 1.25 ppm, and 1.00 ppm. These dilutions were analyzed to establish a consistent detector response across the target concentration range. These solutions were injected into the HPLC system under optimized conditions. The chromatograms from the linearity study and the chromatographic conditions are shown in Figure 12, and the corresponding peak area results are presented in Table 9. The acceptance criteria for the linearity test are as follows [10,13]:Correlation coefficient for the impurity/degradants should be ≥0.990.

### 3.25. Standard Solutions Preparation for Linearity Study

#### 3.25.1. Rivaroxaban Stock Solution (10 ppm)

Transfer 0.1 mL of stock solution rivaroxaban (5000 ppm) into a 50 mL volumetric flask. Complete the volume to the mark with ACN:DI water (70:30 *v*/*v*), and shake it thoroughly.

#### 3.25.2. Solution of Rivaroxaban (2 ppm)

Transfer 5.0 mL of a 10 ppm rivaroxaban stock into a 25 mL volumetric flask. Fill to a certain volume with ACN:DI water (70:30, *v*/*v*) and mix well.

#### 3.25.3. Stock Solution of Rivaroxaban (1.75 ppm)

Pipette 4.38 mL from a 10 ppm rivaroxaban stock into a 25 mL volumetric flask. Dilute with ACN:DI water (70:30, *v*/*v*), and shake until homogeneous.

#### 3.25.4. Stock Solution of Rivaroxaban (1.5 ppm)

Transfer 3.75 mL of the 10 ppm stock solution into a 25 mL volumetric flask. Top off with ACN:DI water (70:30, *v*/*v*), and mix thoroughly.

#### 3.25.5. Stock Solution of Rivaroxaban (1.25 ppm)

Dispense 3.13 mL of the 10 ppm rivaroxaban stock into a 25 mL volumetric flask, dilute with ACN:DI water (70:30, *v*/*v*), and mix.

#### 3.25.6. Stock Solution of Rivaroxaban (1 ppm)

Add 2.5 mL of the 10 ppm rivaroxaban stock to a 25 mL volumetric flask. Complete the volume with ACN:DI water (70:30, *v*/*v*), and shake thoroughly.

### 3.26. Accuracy for Determination of Rivaroxaban Impurities and Degradants

To assess the accuracy of the analytical method in detecting impurities and degradation products, a 5000 ppm rivaroxaban stock solution was diluted to obtain test samples at concentrations of 2 ppm, 1.5 ppm, and 1.0 ppm. Each solution was analyzed using the HPLC system, and the recovery rates for both impurities and degradants were calculated. These calculations were based on the linear regression model for the impurity/degradant response, represented by the equation *y =* 41.91*x* + 8.7932, as illustrated in Figure 13. The peak area values corresponding to each concentration are compiled in Table 9.
$$\mathrm{Percentage}\;\mathrm{Recovery}=\frac{Csample}{Cstandard}\times 100$$where ***Csample*** refers to the concentration determined from the linear regression equation, while ***Cstandard*** denotes the actual, known concentration of the prepared sample.

For the accuracy assessment, triplicate injections were performed for each concentration level (2 ppm, 1.5 ppm, and 1.0 ppm) using the HPLC system. The chromatograms obtained during this analysis, along with the experimental parameters, are presented in Figure 14. According to method validation criteria, the acceptable recovery range is 95% to 105% of the theoretical value. Meeting this criterion confirms that the method provides consistent and precise quantification of both the active pharmaceutical ingredient and its associated impurities or degradation products across multiple concentration levels.

### 3.27. Solutions Preparation for Accuracy Study

#### 3.27.1. Rivaroxaban Stock Solution (10 ppm)

Accurately transfer 2.0 mL of a 100 ppm stock solution into a 100 mL volumetric flask. Dilute to volume with a 70:30 (*v*/*v*) mixture of acetonitrile and deionized water (ACN:DI), and mix thoroughly to ensure uniformity.

#### 3.27.2. Stock Solution of Rivaroxaban (2 ppm)

Pipette 10 mL of the 10 ppm rivaroxaban stock solution into a 50 mL volumetric flask. Dilute to the calibration mark with the same ACN:DI solvent mixture, and mix thoroughly.

#### 3.27.3. Stock Solution of Rivaroxaban (1.5 ppm)

Transfer 7.5 mL of the 10 ppm stock into a 50 mL volumetric flask. Bring the volume up to the mark with the ACN:DI mixture, and mix until homogenous.

#### 3.27.4. Stock Solution of Rivaroxaban (1 ppm)

M5.0 mL of the 10 ppm rivaroxaban solution, and dilute it to 50 mL using the ACN:DI (70:30 *v*/*v*) mixture. Shake well to ensure complete mixing.

### 3.28. Method Precision

Precision in analytical testing signifies the degree to which repeated measurements of the same sample yield similar results under defined conditions, thereby demonstrating the method’s reliability and consistency. In accordance with the International Council for Harmonisation (ICH) guidelines, precision is evaluated at three distinct levels, each reflecting different operational contexts [10]:**Repeatability**: this level examines result consistency when the same analyst conducts multiple measurements of a single sample using the same instrument and protocol over a brief time span;**Intermediate Precision**: also referred to as intra-laboratory precision, it captures variations that may arise from different analysts, instruments, or days within the same laboratory;**Reproducibility**: this broader assessment evaluates the method’s precision across multiple laboratories, determining its performance in diverse testing environments.

Together, these categories provide a comprehensive evaluation of the method’s ability to produce reliable, reproducible data across varying conditions.

### 3.29. Repeatability (Method Precision)

Repeatability examines the consistency of the analytical method when performed by the same analyst, using identical equipment and conditions, over a short time frame. It specifically evaluates variability resulting from sample preparation and handling under controlled conditions, providing insight into the method’s ability to deliver precise results with minimal operational deviation. Regulatory guidelines set strict limits for method precision, typically requiring that the % relative standard deviation (% RSD) of the peak areas not exceed 2%. To assess repeatability for impurities and degradants, six independently prepared rivaroxaban samples (1.5 ppm) were generated from a 5000 ppm stock solution. Each solution was analyzed under the validated HPLC method, and the chromatographic profiles are presented in Figure 15. The resulting peak areas were used to calculate the % RSD, with the acceptance limit set at not more than 2% for impurities and degradant peaks [10,15].

### 3.30. Injection Precision

Injection precision assesses the reliability of the analytical method by measuring variability arising from the instrumentation, including potential inconsistencies from the injector, column, detector, and data processing system during sample introduction. According to regulatory standards, the relative standard deviation (% RSD) of the peak area for rivaroxaban should remain within a maximum limit of 2%. To evaluate this parameter for impurities and degradants, a single 1.5 ppm rivaroxaban solution was prepared using a 5000 ppm stock as the source. This solution was then injected six consecutive times into the HPLC system under the optimized chromatographic conditions, as illustrated in Figure 16. The % RSD of the resulting peak areas was computed to verify compliance with the precision criteria. The established threshold dictates that % RSD for peak areas should not exceed 2% for acceptable precision [10].

### 3.31. Intermediate Precision (Method Robustness)

The intermediate precision of the developed rivaroxaban analytical method was thoroughly assessed to confirm its reproducibility under varied testing conditions. This evaluation involved analyzing the method’s performance across multiple variables, including changes in analyst, analysis date, HPLC system, and chromatographic column. The goal was to ensure the method’s robustness and reliability in routine laboratory environments, regardless of operator or equipment variability. In line with the acceptance criteria for intermediate precision, the % RSD of the peak area for rivaroxaban raw material should not exceed 2.0%. To evaluate the reproducibility of the analytical method under varying conditions, six individual 1.5 ppm rivaroxaban samples were prepared and analyzed. These samples were injected using different HPLC instruments and column setups, applying the same validated method across multiple days and analysts. This approach ensured that the method maintains consistent performance regardless of equipment or operator. The chromatographic profiles and experimental conditions are depicted in Figure 17, while the peak area data obtained from the analysis are summarized in Table 13. The detailed procedure used for preparing the intermediate precision samples, along with the specific parameters of the developed method, are outlined below [4].
**HPLC:**1100 Series HPLC System with MWD (UV/VIS Detector), Agilent Technologies;**Separation Mode:**Isocratic (Reversed-Phase Separation);**Column:**Water XTERRA RP-18 (4.6 × 250 mm, 5 µm);**Mobile Phase:**Solvent A: 25 mM Potassium Phosphate Monobasic buffer 2.9; Solvent B: 100% can;**Solvent Strength:**(70:30 *v*/*v*) Buffer: Can;**Absorbance:**249 nm;**Flow Rate:**1.0 mL/min;**Injection Volume:**15 µL;**Column Temperature:**Ambient;**Run Time:**18 min;

### 3.32. Limit of Detection (LOD)

The limit of detection (LOD) was evaluated to determine the minimum detectable concentration of the analyte that produces a distinguishable response from baseline noise, even if not quantified with exact precision. Rivaroxaban served as the representative compound for estimating impurity and degradant sensitivity [4,10]. According to regulatory standards, the LOD was established based on a signal-to-noise ratio, with a threshold criterion of at least 3:1.

#### Rivaroxaban Stock Solution (10 ppm)

Pipette 0.1 mL of the 5000 ppm rivaroxaban stock solution into a 50 mL volumetric flask. Dilute to volume with a 70:30 (*v*/*v*) mixture of acetonitrile and deionized water, and mix thoroughly to ensure homogeneity.

### 3.33. Limit of Quantitation (LOQ)

To establish the limit of quantitation (LOQ) for rivaroxaban, a dilution series was generated from a 5000 ppm stock solution, and corresponding signal-to-noise ratios were evaluated. Preliminary findings, summarized in Appendix A, indicated that 1.0 ppm met the required acceptance criteria, exhibiting a signal-to-noise ratio of ≥10 and a relative standard deviation (RSD) of ≤10%. To confirm this result, a 1.0 ppm rivaroxaban solution was injected into the HPLC system ten consecutive times. The chromatographic profiles, illustrated in Appendix A, were analyzed to assess both precision and signal integrity [4]. Supporting data, as detailed in Appendix A, reaffirmed that the signal-to-noise ratio exceeded the regulatory threshold, and the %RSD remained within acceptable limits, validating 1.0 ppm as the LOQ for this analytical method [10].

## 4. Conclusions

A novel and rigorously validated reversed-phase high-performance liquid chromatography (RP-HPLC) method was established for the precise quantification of rivaroxaban and its associated impurities in both active pharmaceutical ingredients and finished dosage forms. The method employs isocratic elution using a mobile phase consisting of 25 mM potassium phosphate monobasic buffer (adjusted to pH 2.9) and acetonitrile in a 70:30 volume ratio, operated at a constant flow rate of 1.0 mL/min. Chromatographic separation was carried out on a Thermo Hypersil ODS C18 analytical column (4.6 × 250 mm, 5 µm particle size), interfaced with an Agilent 1100 Series HPLC system equipped with a diode array detector (DAD) for multi-wavelength detection. An injection volume of 15 µL was utilized, with detection carried out at 249 nm under room temperature conditions. This method demonstrates reliable, accurate, and efficient analysis of rivaroxaban in its raw and formulated forms. System suitability test (SST) results confirmed that all chromatographic peaks were fully resolved, with a tailing factor approaching unity, indicating optimal peak symmetry and high column efficiency. The theoretical plate count exceeded the recommended threshold, reflecting exceptional column performance and effective separation. The method exhibited outstanding specificity, with no interference from excipients, degradation products, or other impurities. Stability studies demonstrated no peak loss, degradation, or emergence of additional peaks between the first and final injections, ensuring the stability of rivaroxaban in the prepared solutions throughout the analysis period. Additionally, the peak purity index exceeded the predefined threshold, verifying that the rivaroxaban peak was spectrally pure and untainted by overlapping signals or co-eluting degradation products. The method displayed excellent linearity across the validated concentration range, with a correlation coefficient (R^2^) of 0.9986, indicating a robust linear relationship between concentration and peak response. The method’s limit of detection (LOD) was established at 0.3 ppm, corresponding to a signal-to-noise ratio of 4.0, which surpasses the minimum regulatory requirement of ≥3. The limit of quantitation (LOQ) was calculated to be 1 ppm, ensuring precise and reproducible quantification even at lower analyte concentrations. Comprehensive validation was conducted in compliance with ICH, USP, and FDA guidelines (ICH Q2(R1)), assessing key parameters including system suitability, specificity, linearity, accuracy, precision, robustness, solution stability, LOD, and LOQ. The method demonstrated high accuracy, with recovery values within the acceptable range of 98.0–105.0%, and excellent precision, with %RSD values consistently below 2.0%, confirming the reproducibility of results. Robustness testing confirmed the method’s reliability under deliberate variations in chromatographic conditions such as mobile phase composition, flow rate, column temperature, and detection wavelength. This validated HPLC method, designed to indicate stability, is highly effective for routine quality assessment, stability evaluation, and analytical testing of rivaroxaban in both bulk drug substances and final pharmaceutical products. Its straightforward execution, strong sensitivity, high analytical efficiency, and dependable robustness make it a preferred option for regulatory-compliant testing in pharmaceutical quality-control environments. Additionally, its capacity to detect potential degradants and impurities further enhances its applicability in stability testing and formulation development.

## Figures and Tables

**Figure 1 ijms-26-04744-f001:**
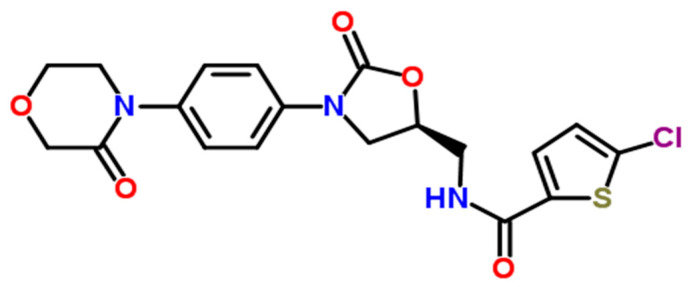
Rivaroxaban chemical structure: a direct oral anticoagulant derived from an oxazolidinone core [4].

**Figure 2 ijms-26-04744-f002:**
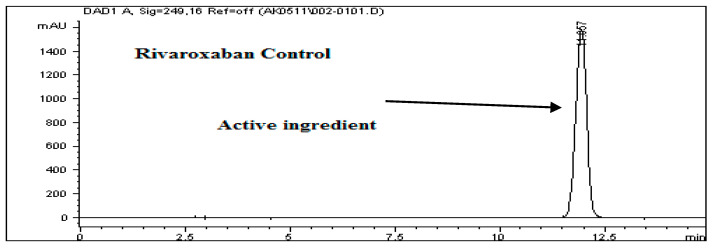
Chromatographic profile of the rivaroxaban control solution (700 ppm) employed in the degradation experiments.

**Figure 3 ijms-26-04744-f003:**
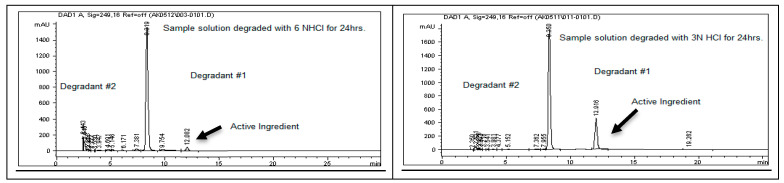
Representative chromatograms of rivaroxaban samples subjected to acid-induced degradation at varying HCl concentrations.

**Figure 4 ijms-26-04744-f004:**
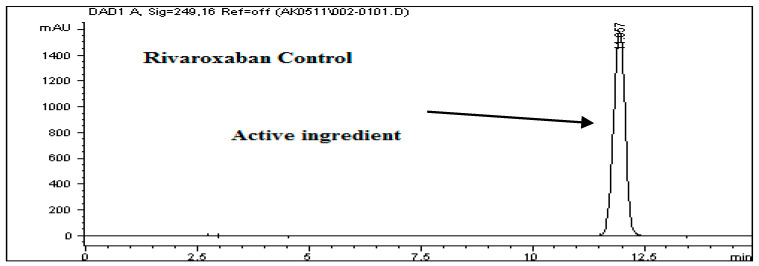
Baseline chromatogram of undegraded rivaroxaban control solution (700 ppm) used for forced degradation comparisons.

**Figure 5 ijms-26-04744-f005:**
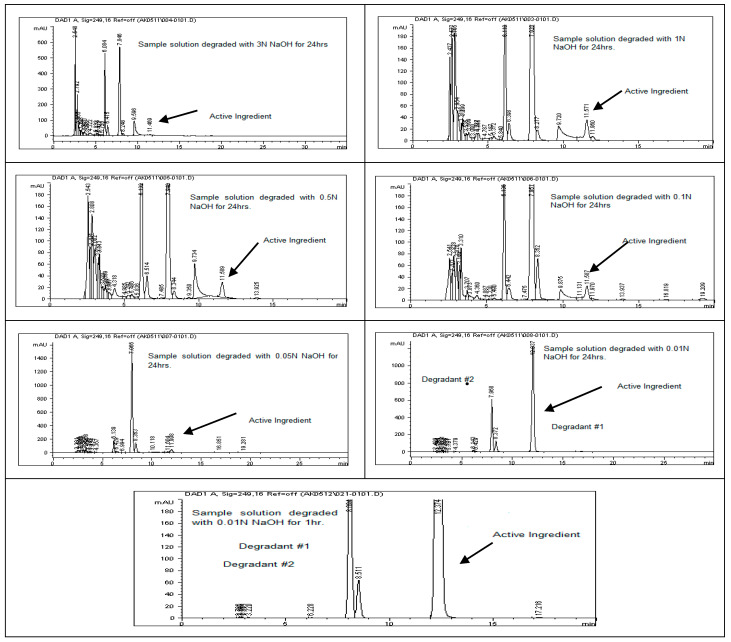
Chromatographic profiles of rivaroxaban samples subjected to alkaline hydrolysis using different concentrations of sodium hydroxide (NaOH).

**Figure 6 ijms-26-04744-f006:**
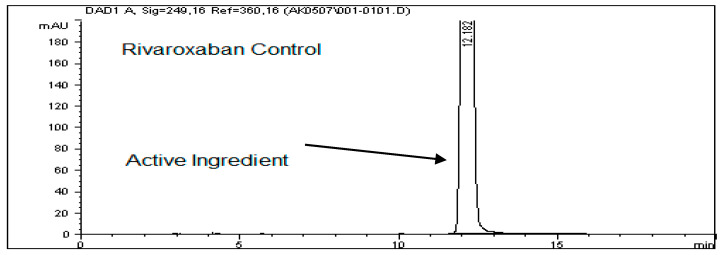
Baseline chromatogram of undegraded rivaroxaban (700 ppm) control sample used for comparative degradation analysis.

**Figure 7 ijms-26-04744-f007:**
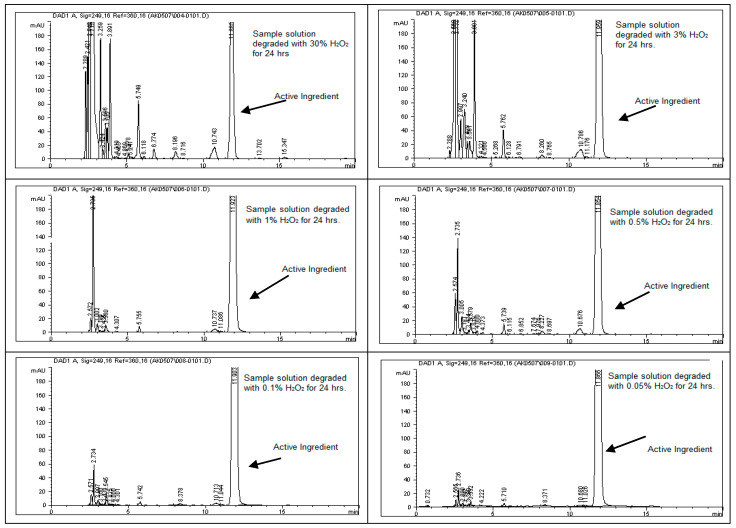
Chromatograms of rivaroxaban solutions exposed to oxidative degradation using varying Concentrations of hydrogen peroxide (H_2_O_2_).

**Figure 8 ijms-26-04744-f008:**
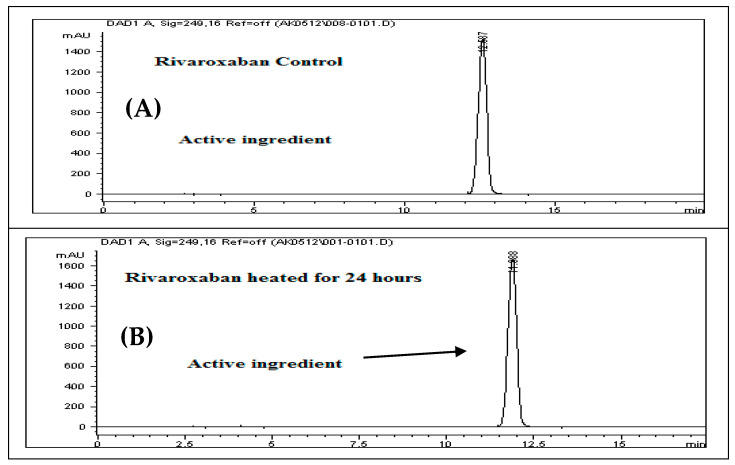
(**A**) Chromatogram of the rivaroxaban control solution (700 ppm) utilized in the degradation assessment; (**B**) chromatogram of rivaroxaban following thermal degradation after 24-h exposure at 75 °C.

**Figure 9 ijms-26-04744-f009:**
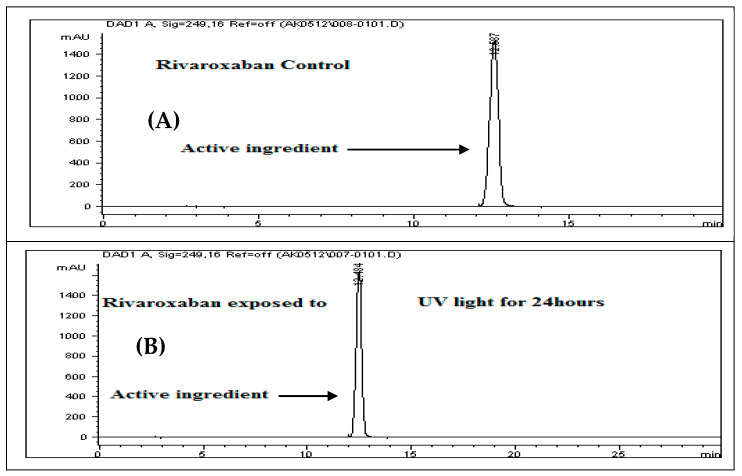
(**A**) Chromatogram of the untreated rivaroxaban control sample (700 ppm) utilized in the photolysis study; (**B**) chromatogram illustrating the degradation profile of rivaroxaban following ultraviolet (UV) light exposure during photolytic stress testing.

**Figure 10 ijms-26-04744-f010:**
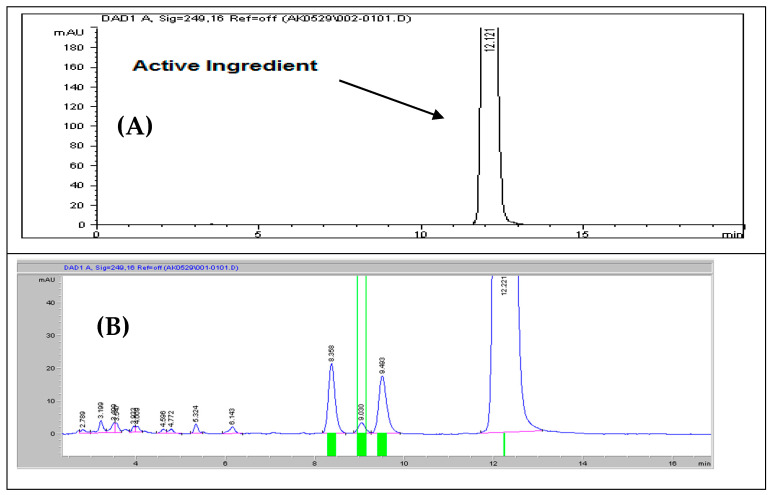
(**A**) Chromatogram of the rivaroxaban control sample (700 ppm) used as a reference in the forced degradation experiments; (**B**) magnified view of the chromatogram showcasing the effective resolution of multiple degradation products obtained under optimized mixed-stress degradation conditions.

**Figure 11 ijms-26-04744-f011:**
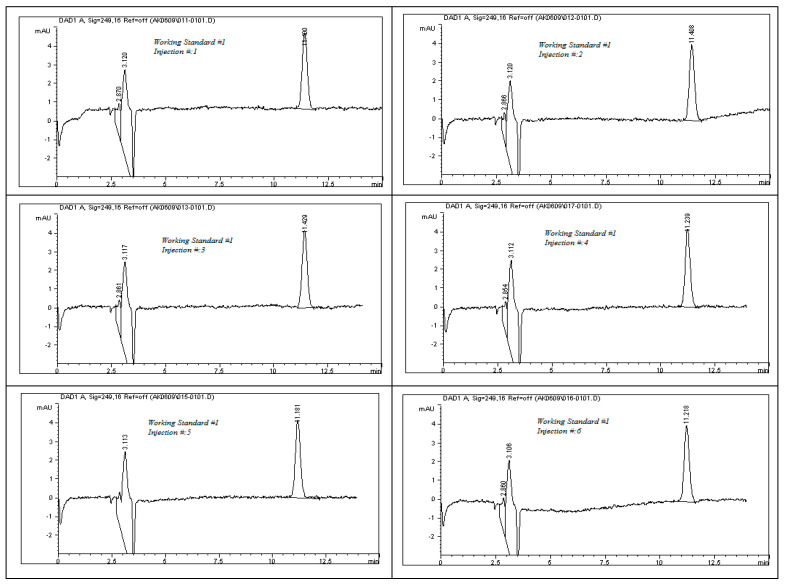
Representative chromatograms of rivaroxaban working standards #1 and #2 utilized for evaluating system suitability parameters, including peak symmetry, retention time consistency, and theoretical plate count.

**Figure 12 ijms-26-04744-f012:**
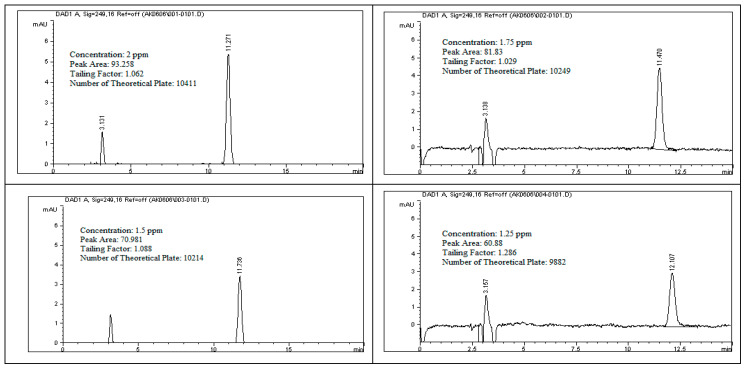
Chromatograms obtained during the linearity assessment of rivaroxaban impurities across multiple concentration levels.

**Figure 13 ijms-26-04744-f013:**
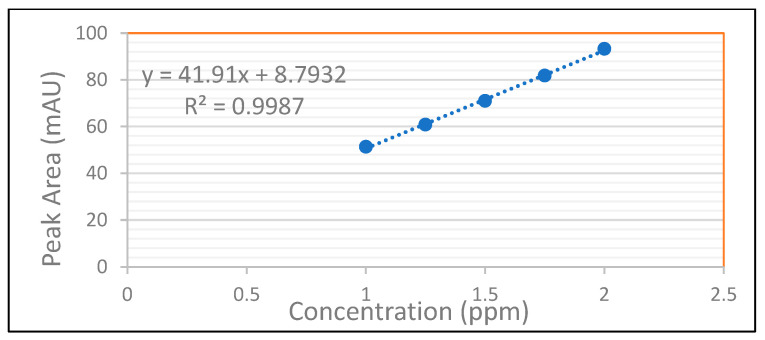
Plot of the peak area versus concentration for rivaroxaban in the linearity study of impurities and degradants. This graph illustrates the method’s linear response across the tested concentration range for accurate quantification.

**Figure 14 ijms-26-04744-f014:**
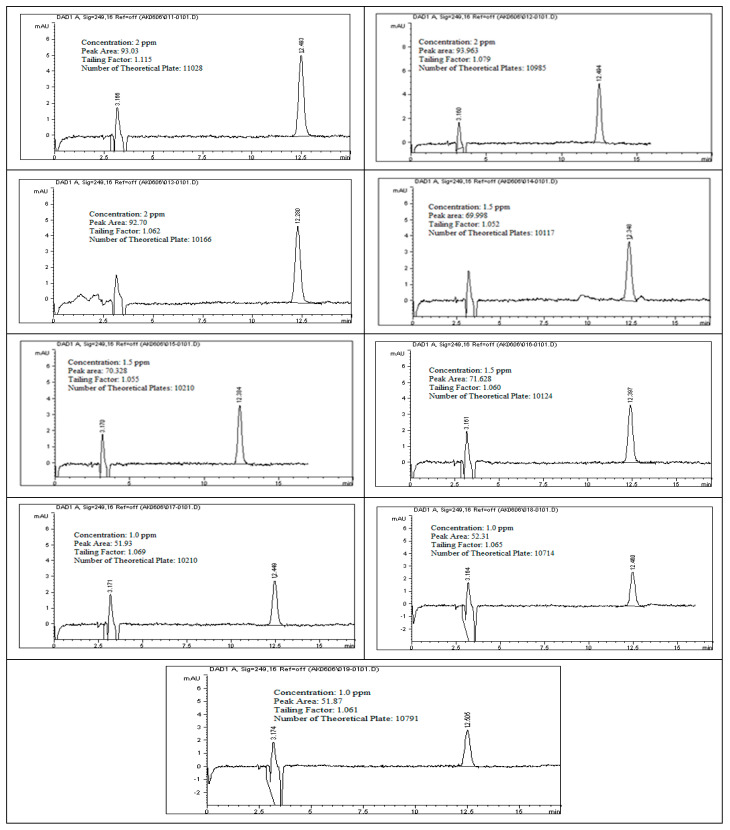
Chromatograms from the accuracy study of rivaroxaban (2 ppm, 1.5 ppm, and 1.0 ppm) impurities and degradants. The analysis demonstrates the method’s ability to accurately quantify impurities and degradants at the specified concentration.

**Figure 15 ijms-26-04744-f015:**
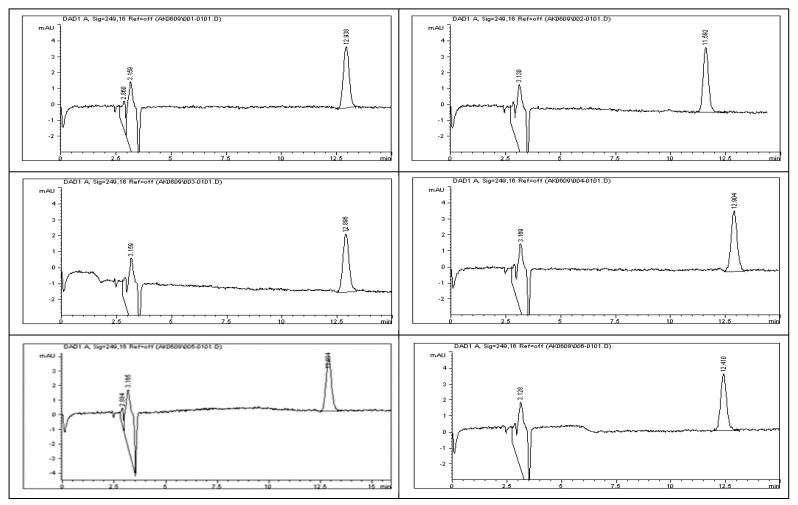
Chromatograms from the method precision study for rivaroxaban at 1.5 ppm, showing impurities and degradants. These chromatograms demonstrate the reproducibility and consistency of the method under the specified conditions.

**Figure 16 ijms-26-04744-f016:**
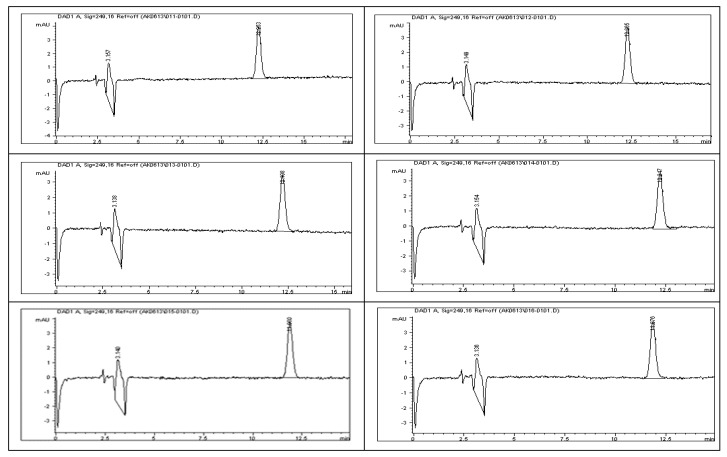
Chromatograms from the injection precision study for rivaroxaban at 1.5 ppm, highlighting impurities and degradants. These chromatograms illustrate the precision and consistency of injections under the specified conditions.

**Figure 17 ijms-26-04744-f017:**
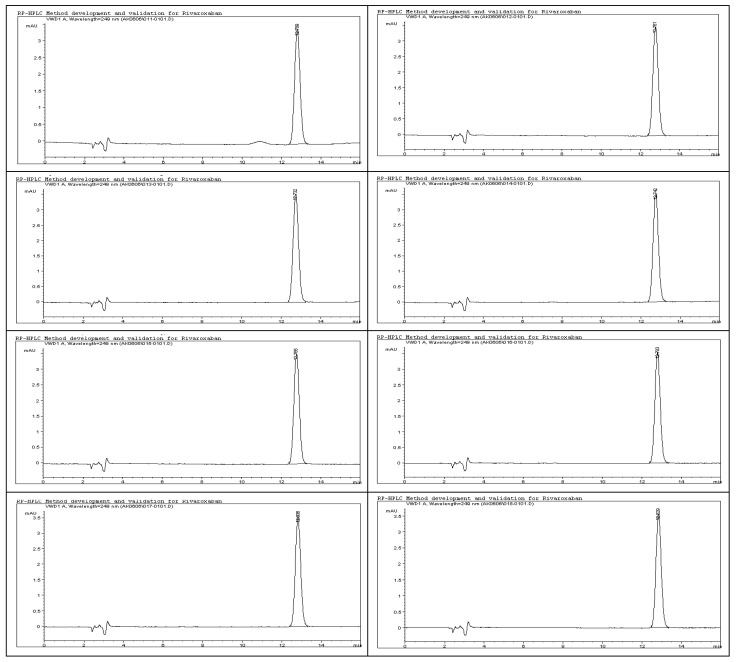
Chromatograms from the intermediate precision study for rivaroxaban at 1.5 ppm, showing impurities and degradants. These chromatograms demonstrate the consistency of the method across different instruments and testing conditions.

**Table 1 ijms-26-04744-t001:** Optimized chromatographic parameters for rivaroxaban analysis [4].

Parameters	Conditions
Column	C18 (4.6 × 250 mm, 5 µm) Thermo ODS Hypersil
Mobile Phase	ACN: BUFFER (25 mM potassium phosphate buffer monobasic pH 2.9) (30:70)
RT (retention time)	12.1 ± 0.235 min
Flow Rate	1 mL/min
Injector	15 µL loop
Wavelength	249 nm
Column Temperature	Ambient

**Table 2 ijms-26-04744-t002:** Overview of rivaroxaban degradation behavior in acidic conditions using optimized HPLC parameters.

Stress Condition	Exposed Time	Temperature (°C)	Solution Color	Concentration (ppm)	Peak Area	Percent Degradation
Control	None	None	None	700	29,460.8	-
6 N HCl	24 h	75 °C	Clear	700	622.8	97.89
3 N HCl	24 h	75 °C	Clear	700	6562.2	77.73
1 N HCl	24 h.	75 °C	Clear	700	14,611.0	50.41
0.5 N HCl	24 h	75 °C	Clear	700	18,169.0	38.33
0.1 N HCl	24 h	75 °C	Clear	700	26,862.4	8.82
0.05 N HCl	24 h	75 °C	Clear	700	25,466.0	13.56
0.01 N HCl	24 h	75 °C	Clear	700	26,868.3	8.8

**Table 3 ijms-26-04744-t003:** Evaluation of rivaroxaban stability under alkaline conditions using optimized HPLC method.

Stress condition	Exposed Time	Temperature (°C)	Solution Color	Concentration (ppm)	Peak Area	Percent Degradation
Control	None	None	None	700	29,460.8	-
3 N NaOH	24 h	75 °C	Yellowish	700	85.90	99.71
1 N NaOH	24 h	75 °C	Black	700	677.9	97.70
0.5 N NaOH	24 h	75 °C	Black	700	789.60	97.30
0.1 N NaOH	24 h	75 °C	Yellowish	700	980.5	96.70
0.05 N NaOH	24 h	75 °C	Yellowish	700	1120.0	96.20
0.01 N NaOH	24 h	75 °C	Clear	700	17,509.1	40.57
0.01 N NaOH	1 h	75 °C	Clear	700	27,074.5	8.1

**Table 4 ijms-26-04744-t004:** Assessment of oxidative degradation of rivaroxaban using optimized HPLC parameters.

Stress Condition	Exposed Time	Temperature (°C)	Solution Color	Concentration (ppm)	Peak Area	Percent Degradation
Control	None	None	None	700	30,841.7	-
30% H_2_O_2_	24 h	75 °C	Clear	700	4391.50	85.80
3% H_2_O_2_	24 h	75 °C	Clear	700	16,904.0	45.20
1% H_2_O_2_	24 h.	75 °C	Clear	700	17,042.0	44.75
0.5% H_2_O_2_	24 h	75 °C	Clear	700	26,783.0	13.16
0.1% H_2_O_2_	24 h	75 °C	Clear	700	27,838.0	9.74
0.05% H_2_O_2_	24 h	75 °C	Clear	700	28,837.0	6.50

**Table 5 ijms-26-04744-t005:** Evaluation of thermal stress effects on 700 ppm rivaroxaban using optimized HPLC conditions.

Stress Condition	Exposed Time	Temperature (°C)	Color	Peak Area	Percent Degradation
Control	None	None	Clear	29,894.4	-
Heat	24 h	75 °C	Clear	29,595.9	1.0

**Table 6 ijms-26-04744-t006:** Assessment of photolytic degradation of 700 ppm rivaroxaban under validated HPLC parameters.

Stress Condition	Exposed Time	Temperature (°C)	Color	Peak Area	Percent Degradation
Control	None	None	Clear	29,894.4	-
Photolysis	24 h	None	Clear	29,675.3	0.83

**Table 7 ijms-26-04744-t007:** Identified rivaroxaban impurities and unknown degradation products.

No.	Retention Time (min)	Impurity Name	IUPAC Name
1	2.79	G	4-[4-(5-Aminomethyl-2-oxo-oxazolidin-3-yl)-phenyl]-morpholin-3-one
2	3.20	*U	
3	3.50	D	N-{2-oxo-3-[4-(3-oxo-morpholin-4-yl)-phenyl]-oxazolidin-5-ylmethyl}-acetamide
4	4.00	*U	
5	4.59	*U	
6	4.77	*U	
7	5.32	H	5-Chlorothiophene-2-carboxylic acid
8	6.14	C	1,3-Bis-{2-oxo-3-[4-(3-oxo-morpholin-4-yl)-phenyl]-oxazolidin-5-ylmethyl}-urea
9	8.36	E	Thiophene-2-carboxylic acid {2-oxo-3-[4-(3-oxo-morpholin-4-yl)-phenyl]-oxazolidin-5-ylmethyl}-amide
10	9.03	A	{2-[4-(5-{[(5-Chloro-thiophene-2-carbonyl)-amino]-methyl}-2-oxo-oxazolidin-3-yl)-phenylamino]-ethoxy}-acetic acid
11	9.49	F	2-{2-Oxo-3-[4-(3-oxo-morpholin-4-yl)-phenyl]-oxazolidin-5-ylmethyl}-isoindole-1,3-dione

*U: Unknown degradation products.

**Table 8 ijms-26-04744-t008:** Evaluation of system suitability parameters for rivaroxaban using working standard solutions #1 and #2, including retention time, theoretical plates, tailing factor, and peak area reproducibility under optimized chromatographic conditions.

Standard #1 Injection	Retention Time	Peak Area (mAU)	Plate Count	Tailing Factor
1	11.300	68.10	10,928	1.025
2	11.408	67.10	10,850	1.085
3	11.429	68.49	11,275	1.014
4	11.239	68.50	10,489	1.104
5	11.181	68.77	11,082	1.081
6	11.218	68.10	10,582	1.090
Average	11.312	68.18		
STDEV	0.087	0.59		
Retention Time % RSD	0.77			
Peak Area % RSD		0.86		
% Drift	0.37			
**Standard #2 Injection**	**Retention Time**	**Peak Area (mAU)**	**Plate Count**	**Tailing Factor**
1	11.242	68.270	10,909	1.097
2	11.293	68.50	11,008	1.050
Average	11.27	68.39		
STDEV	0.0361	0.163		
Retention Time % RSD	0.320			
Peak Area % RSD		0.24		

**Table 9 ijms-26-04744-t009:** Calibration curve data demonstrating the linear relationship between concentration and peak response for rivaroxaban impurities and degradation products.

Sample #	Concentration (ppm)	Retention Time (min)	Peak Area (mAu)
1	2.0	11.271	93.258
2	1.75	11.470	81.83
3	1.5	11.736	70.981
4	1.25	12.107	60.88
5	1.0	12.500	51.345

**Table 10 ijms-26-04744-t010:** Accuracy result for impurities and degradants.

Concentration (ppm)	Peak Area	Retention Time (min)	Percent Recovery	Average Percent Recovery	%RSD
2	93.030	12.490	100.6	100.8	0.77
	93.963	12.494	101.7		
	92.70	12.280	100.2		
1.5	69.998	12.393	97.5	98.6	1.34
	70.328	12.394	98.3		
	71.628	12.397	100.1		
1	51.930	12.449	103.1	103.4	0.55
	52.310	12.406	104.0		
	51.870	12.505	103.0		

**Table 11 ijms-26-04744-t011:** Method precision results for rivaroxaban impurities and degradants.

Preparation Sample	Concentration (ppm)	Retention Time (min)	Peak Area	Tailing Factor	Number of Theoretical Plates
1	1.5	12.253	71.26	1.099	11,552
2	1.5	12.265	71.1	1.119	11,012
3	1.5	12.198	71.1	1.199	10,467
4	1.5	12.247	72.79	1.021	11,226
5	1.5	11.980	72.83	1.050	11,186
6	1.5	11.976	72.90	1.166	10,982
Average Peak Area		72.0			
Standard Deviation		0.93			
%RSD		1.29			

**Table 12 ijms-26-04744-t012:** Injection precision results for rivaroxaban impurities and degradants.

Sample Injection	Concentration (ppm)	Peak Area	Tailing Factor	Retention Time (min)	Number of Theoretical Plates
1	1.5	74.10	1.065	12.253	9242
2	1.5	73.90	1.046	12.265	9260
3	1.5	71.20	1.091	12.198	9361
4	1.5	73.70	1.020	12.247	9441
5	1.5	72.90	1.018	11.980	9321
6	1.5	73.50	1.105	11.976	9291
Average		73.22			
Standard Deviation		1.07			
%RSD		1.46			

**Table 13 ijms-26-04744-t013:** Intermediate precision results for rivaroxaban impurities and degradants.

Sample #	Concentration (ppm)	Peak Area	Retention Time (min)	Tailing Factor	Number of Theoretical Plates
1	1.5	67.81	12.78	0.93	9848
2	1.5	67.78	12.78	0.92	9963
3	1.5	68.04	12.73	0.94	9919
4	1.5	67.87	12.74	0.94	10,096
5	1.5	67.43	12.76	0.94	10,299
6	1.5	67.62	12.78	0.94	10,328
7	1.5	67.49	12.8	0.95	10,366
8	1.5	67.17	12.82	0.94	10,576
9	1.5	67.63	12.84	0.94	10,595
10	1.5	67.4	12.82	0.93	10,750
Average		67.62	12.79		
Standard Deviation		0.259	0.036		
%RSD		0.38	0.28		

## Data Availability

All relevant data supporting the findings of this study are included within the article and its Appendix A. Further inquiries can be directed to the corresponding author.

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
