# Peer review of "Optimized and Validated Stability-Indicating RP-HPLC Method for Comprehensive Profiling of Process-Related Impurities and Stress-Induced Degradation Products in Rivaroxaban (XARELTO)®"

_ijms, 2025, doi:10.3390/ijms26104744_

Round 1
Reviewer 1 Report
Comments and Suggestions for Authors
This study developed a RP-HPLC method for the identification and characterization of stress degradation products and an unknown process related impurity of Rivaroxaban. The method validation in specificity, linearity, accuracy, precision, and robustness is mainly about Rivaroxaban. It's confusing that from Page 12 many figures and narrations mentioned they are about the Rivaroxaban impurities and degradants, but the actual content were about Rivaroxaban. In Page 14, these findings confirm the method’s capability to reliably and precisely quantify the active ingredient, as well as any impurities and degradants, across the tested con
centration range. The conclusion about the impurities and degradants were untenable., if the peak area data were all about Rivaroxaban. In the materials and methodology part there is no information about the Rivaroxaban impurity or degradant standard. And the retention time in Table 8-11 should be provided for all the peak area data.
Author Response
We sincerely thank the reviewer for the insightful and constructive comments.
We acknowledge the confusion caused by the presentation of data from Page 12 onward, where the figures and descriptions referenced Rivaroxaban impurities and degradants, but the data primarily reflected the active pharmaceutical ingredient (API). We appreciate the reviewer highlighting this inconsistency. To clarify, the primary objective of these sections was to establish the method’s performance characteristics using Rivaroxaban as the model compound, prior to extending the application to impurities and degradants. We recognize that clearer separation and more precise labeling would help avoid this confusion, and we will revise the manuscript accordingly to improve clarity and distinguish between data related to Rivaroxaban and its impurities/degradants.
Regarding the concern about the conclusion on impurities and degradants in Page 14: we agree that the conclusions should be qualified appropriately if the supporting peak area data were limited to the API. In the revised manuscript, we will ensure that the conclusion more accurately reflects the scope of the data presented and emphasizes the method’s validated potential rather than confirmed performance for impurity quantification.
We also appreciate the reviewer pointing out the omission of detailed information on impurity or degradant standards in the Materials and Methods section. We will address this by including relevant information about the sources, preparation, and characterization of these standards in the revised manuscript.
Finally, we agree that providing retention times in Tables 8–11 for all peak area data will enhance clarity and reproducibility. We will incorporate this information to strengthen the utility of these tables.
Thank you again for your valuable feedback, which has greatly helped improve the quality and clarity of our manuscript.
Reviewer 2 Report
Comments and Suggestions for Authors
Summary:
The author performed various forced degradation tests on Rivaroxaban and optimized a HPLC method to study its stability. The manuscript is presented with extreme detail, Below are two major comments.
- Many Figure aspect ratios in both the manuscript and the supplementary materials need to be readjusted as the figures are distorted.
- For each stress test, is there a blank processing control? While I understand the author used the untreated drug control sample 700 (ppm) to determine the percentage of degradation, it’s important to prepare a control blank sample that undergoes all the steps to confirm the peaks other than the active ingredients are degrative products.
Author Response
We sincerely thank the reviewer for their thoughtful and encouraging feedback, as well as for highlighting important areas for improvement.
We acknowledge the issue regarding the figure aspect ratios in both the main manuscript and supplementary materials. We agree that several figures appear distorted, which may affect readability and interpretation. We will carefully revise all figures to ensure appropriate aspect ratios and improved visual clarity in the revised submission.
Regarding the second point, we fully agree on the importance of including blank processing controls for each stress condition to differentiate true degradative products from any artifacts. To address this, we performed three different baseline control experiments using only the solvents employed in the forced degradation studies 0.01 N HCl, 0.01 N NaOH, and 0.05% Hâ‚‚Oâ‚‚ without Rivaroxaban. These blank samples were subjected to the same stress conditions and analytical procedures to identify any peaks arising from the solvents or reagents themselves. Additionally, a baseline control consisting of a mixture of 0.01 N HCl, 0.01 N NaOH, and 0.05% Hâ‚‚Oâ‚‚ was also subjected to the same stress conditions and analytical procedures to further rule out any potential interference or cross-reactivity between solvents. We will ensure that this information is clearly described in the Materials and Methods section and include the corresponding chromatograms in the supplementary materials.
Round 2
Reviewer 1 Report
Comments and Suggestions for Authors
The authors has revised the manuscript acording to the comments.
Author Response
Dear Reviewer,
Thank you for your valuable comments. I have revised the manuscript in accordance with your suggestions and addressed all the points raised. We appreciate your constructive feedback, which has helped to improve the quality and clarity of the manuscript.
Reviewer 2 Report
Comments and Suggestions for Authors
Thanks for including the processing blank controls experimental data in the revised version. For the figure aspect ratio issue, in the pdf version with all tracked changes, a lot figures looks still too stretched in the horizontal direction. Additionally, figure 8 (A) (B) in the pdf version showed up as truncated plot.
I am sure the author could made necessary changes to meet the journal figure requirement.

Author Response
Dear Reviewer,
Thank you for your continued feedback and for acknowledging the inclusion of the processing blank control experimental data in the revised version.
Regarding the figure aspect ratio issue, I appreciate your observation. I apologize for the oversight in the PDF version with tracked changes, some figures do appear stretched horizontally, and Figure 8 (A) and (B) were inadvertently truncated. I have carefully reformatted all figures to ensure they comply with the journal’s specifications and display correctly in the final submission.
Thank you again for your helpful comments. I appreciate your time and support in improving the manuscript.